# Cytotoxic Effects of Diterpenoid Alkaloids Against Human Cancer Cells

**DOI:** 10.3390/molecules24122317

**Published:** 2019-06-22

**Authors:** Koji Wada, Hiroshi Yamashita

**Affiliations:** Department of Medicinal Chemistry, Faculty of Pharmaceutical Sciences, Hokkaido University of Science, 4-1, Maeda 7-jo 15-choume, Teine-ku, Sapporo 006-8590, Japan; yama@hus.ac.jp

**Keywords:** diterpenoid alkaloids, cytotoxicity, human tumor cells, lipojesaconitine, delcosine, delpheline, kobusine, pseudokobusine

## Abstract

Diterpenoid alkaloids are isolated from plants of the genera *Aconitum*, *Delphinium*, and *Garrya* (Ranunculaceae) and classified according to their chemical structures as C_18_-, C_19_- or C_20_-diterpenoid alkaloids. The extreme toxicity of certain compounds, e.g., aconitine, has prompted a thorough investigation of how structural features affect their bioactivities. Therefore, natural diterpenoid alkaloids and semi-synthetic alkaloid derivatives were evaluated for cytotoxic effects against human tumor cells [A549 (lung carcinoma), DU145 (prostate carcinoma), MDA-MB-231 (triple-negative breast cancer), MCF-7 (estrogen receptor-positive, HER2-negative breast cancer), KB (identical to cervical carcinoma HeLa derived AV-3 cell line), and multidrug-resistant (MDR) subline KB-VIN]. Among the tested alkaloids, C_19_-diterpenoid (e.g., lipojesaconitine, delcosine and delpheline derivatives) and C_20_-diterpenoid (e.g., kobusine and pseudokobusine derivatives) alkaloids exhibited significant cytotoxic activity and, thus, provide promising new leads for further development as antitumor agents. Notably, several diterpenoid alkaloids were more potent against MDR subline KB-VIN cells than the parental drug-sensitive KB cells.

## 1. Introduction

Cancer therapy mainly involves surgery, chemotherapy, radiation therapy, immunotherapy, monoclonal antibody therapy, and hormone therapy. Chemotherapy generally refers to the use of cytotoxic drugs to treat cancer. Plant alkaloids are one major class of chemotherapeutic drugs [1,2,3,4,5,6,7,8,9]. Chemotherapeutic drugs that affect cell division by preventing the normal functioning of micro-tubules include the vinca alkaloids. 

Numerous diterpenoid alkaloids have been isolated from various *Aconitum*, *Delphinium*, and *Garrya* (Family Ranunculaceae) species and are classified according to their chemical structures as C_18_-, C_19_- or C_20_-diterpenoid alkaloids (Figure 1) [10,11]. The C_19_-diterpenoid alkaloids may be divided into six types: aconitine, lycoctonine, pyro (C_8_=C_15_ or C_15_=O), lactone (δ-valerolactone rather than cyclopentyl C-ring), 7,17-*seco*, and rearranged ones [10,11]. Most of the isolated C_19_-diterpenoid alkaloids are aconitine- and lycoctonine-types and include aconitine, mesaconitine, hypaconitine and jesaconitine, all of which are extremely toxic. The C_20_-diterpenoid alkaloids may be divided into ten types: atisine, denudatine, hetidine, hetisine, vakognavine, napelline, kusnezoline, racemulosine, arcutine, and tricalysiamide [10,11]. Most of the isolated C_20_-diterpenoid alkaloids are atisine-, hetisine-, and napelline-types and include atisine, kobusine, pseudokobusine and lucidusculine, which are far less toxic [12]. 

The pharmacological properties of the C_19_-diterpenoid alkaloids have been studied extensively and reviewed [12]. Aconitine is a toxin that exhibits activity both centrally and peripherally, acting predominantly on the cardiovascular and respiratory systems by preventing the normal closing of sodium channels [12]. This extreme toxicity resulted in the use of *Aconitum* extracts as poisons in hunting and warfare [13], although extracts were also used as traditional medicines by oral and topical routes. For example, the roots of *Aconitum* plants have been used as “bushi”, an herbal drug in some prescriptions of traditional Japanese medicine for the treatment of hypometabolism, dysuria, cardiac weakness, chills, neuralgia, gout, and certain rheumatic diseases [14]. However, proper processing is essential to reduce the content of toxic alkaloids and avoid inadvertent poisoning [15,16,17]. Such obstacles encourage a good understanding of the relationships between structure and cytotoxic activity of aconitine and related compounds before they can be considered for modification and development as chemotherapeutic agents.

Our previous study demonstrated the effects of various naturally occurring and semi-synthetic C_19_- and C_20_-diterpenoid alkaloids on the growth of the A172 human malignant glioma cell line [18]. Antitumor properties and radiation-sensitizing effects of various types of novel derivatives prepared from C_19_- and C_20_-diterpenoid alkaloids were also investigated [19]. Two novel hetisine-type C_20_-diterpenoid derivatives showed significant suppressive effects against the Raji non-Hodgkin’s lymphoma cell line [20]. In addition, the effects of various semi-synthetic novel hetisine-type C_20_-diterpenoid alkaloids on the growth of the A549 human lung cancer cell line were examined and subsequent structure-activity relationships for the antiproliferative effects against A549 cells were considered [21]. Since 2012, several diterpenoid alkaloid components and their derivatives exhibited antiproliferative activity against human tumor cell lines, including A549 (lung carcinoma), DU145 (prostate carcinoma), MDA-MB-231 (estrogen and progesterone receptor-negative & HER2-negative triple-negative breast cancer), MCF-7 (estrogen receptor-positive, HER2-negative breast cancer), KB (identical to cervical carcinoma HeLa derived AV-3 cell line), and multidrug-resistant (MDR) subline KB-VIN [P-glycoprotein (P-gp) overexpressing vincristine-resistant KB subline]. Among such alkaloids, C_19_-diterpenoid (e.g., lipojesaconitine, delpheline, and delcosine derivative) and C_20_-diterpenoid (e.g., kobusine and pseudokobusine derivatives) alkaloids have shown significant antiproliferative activity, as well as provided promising new leads for further development as antitumor agents.

## 2. Antiproliferative Activity of C_19_-Diterpenoid Alkaloid Derivatives

### 2.1. Aconitine-Type C_19_-Diterpenoid Alkaloids

The tested aconitine-type C_19_-diterpenoid alkaloids included 21 natural alkaloids, aconitine (**1**), deoxyaconitine (**2**), jesaconitine (**3**), deoxyjesaconitine (**4**), aljesaconitine A (**5**), secojesaconitine (**6**), mesaconitine (**8**), hypaconitine (**9**), hokbusine A (**10**), 14-anisoyllasianine (**12**), *N*-deethylaljesaconitine A (**13**), aconine (**14**), lipomesaconitine (**15**), lipoaconitine (**16**), lipojesaconitine (**17**), neolinine (**18**), neoline (**19**), 14-benzoylneoline (**20**), isotalatizidine (**21**), karacoline (**22**), and 3-hydroxykaracoline (**23**), isolated from the rhizoma of *Aconitum japonicum* THUNB. subsp. *subcuneatum* (NAKAI) KADOTA [22,23,24,25,26,27,28] (Figure 2). Two synthetic aconitine-type C_19_-diterpenoid alkaloids, 3,15-diacetyljesaconitine (**7**) [26] and 3-acetylmesaconitine (**11**) [29] prepared from secojesaconitine (**6**) and mesaconitine (**8**), respectively (Figure 2), were also tested. 

Eighteen of the 23 tested aconitine-type C_19_-diterpenoid alkaloids, both natural alkaloids (**1**~**6**, **8**~**10**, **12**~**14**, **18**~**23**) and synthetic analogs (**7** and **11**), were inactive (IC_50_ > 20 or 40 μM) [27,28,30] (Table 1). Three natural diterpenoid alkaloids (**15**~**17**) exhibited cytotoxic activity against five human tumor cell lines (A549, MDA-MB-231, MCF-7, KB, and MDR KB subline KB-VIN) (Table 1). Lipojesaconitine (**17**) showed significant cytotoxicity against four tested cell lines with IC_50_ values of 6.0 to 7.3 μM, but weak cytotoxicity against KB-VIN (IC_50_ = 18.6 μM) [28]. Lipomesaconitine (**15**) showed moderate cytotoxicity against the KB cell line (IC_50_ = 9.9 μM), but weak cytotoxicity against the other four human tumor cell lines (IC_50_ =17.2 ~ 21.5 μM) [27]. Lipoaconitine (**16**) was weakly cytotoxic (IC_50_ = 13.7 ~ 20.3 μM) against all five human tumor cell lines [28]. Based on the results, the fatty acid ester at C-8 and the anisoyl group at C-14 found in **17** may be important to the cytotoxic activity of aconitine-type C_19_-diterpenoid alkaloids.

### 2.2. Lycoctonine-Type (7,8-diol) C_19_-Diterpenoid Alkaloids

The tested lycoctonine-type (7,8-diol) C_19_-diterpenoid alkaloid group included 12 natural alkaloids, namely nevadensine (**24**), *N*-deethylnevadensine (**25**), and virescenine (**27**), purified from rhizoma of *Aconitum japonicum* subsp. *subcuneatum* [27], and 18-methoxygadesine (**26**), delphinifoline (**28**), delcosine (**34**), 14-acetyldelcosine (**34–43**), and 14-acetylbrowniine (**35**), purified from root of *Aconitum yesoense* var. *macroyesoense* (NAKAI) TAMURA [31,32,33,34], and andersonidine (**30**), pacifiline (**31**), pacifinine (**32**), and pacifidine (**33**), purified from seeds of *Delphinium elatum* cv. Pacific Giant [35] (Figure 3). The remaining tested C_19_-diterpenoid alkaloids from this subtype were synthetic alkaloids, *N*-deethyldelsoline (**29**) [18], 1-acetyldelcosine (**34-1**) [36], 1,14-diacetyldelcosine (**34-2**) [37], 1-(4-trifluoromethylbenzoyl)delcosine (**34-24**) [30], delsoline (**34-42**) [37], 1,14-di-(4-nitrobenzoyl)-delcosine (**34-45**) [30], 14-acetyl-1-(4-nitrobenzoyl)delcosine (**34-46**) [30], and 1-acyl or 1,14-diacyldelcosine derivatives (**34-3**~**34-23**, **34-25**~**34-41**, **34-44**, and **34-47**) [38], prepared from delcosine (**34**) or delsoline (**34-42**) (Figure 3). These 42 C_19_-diterpenoid alkaloids were evaluated for antiproliferative activity against four to five human tumor cell lines (A549, DU145, MDA-MB-231, MCF-7, KB, and KB-VIN) [30,38] (Table 2). Several tested lycoctonine-type (7,8-diol) C_19_-diterpenoid alkaloids, both natural alkaloids (**24**~**28**, **30**~**33**) and a synthetic alkaloid (**29**), were inactive (IC_50_ > 20 or 40 μM). All tested delcosine derivatives that contain an acetyl or methoxy group, both natural alkaloids (**34**, **34-43**, **35**) and synthetic analogs (**34-1**, **34-2**, **34-42**), were inactive (IC_50_ > 20 μM). However, acylation, except with an acetyl group, of the C-1 and/or C-14 hydroxy group of **34** led to various degrees of antiproliferative activity. Among the C-1 esterified alkaloids, the synthetic derivatives **34-6**, **34-8**, **34-10**, and **34-18** exhibited significant potency against all cell lines (average IC_50_ 9.3, 5.3, 5.0, and 6.9 µM, respectively). Also, alkaloids **34-3**, **34-16**, **34-17**, **34-21**, **34-25**, **34-27**, **34-31**, **34-32**, **34-38**, and **34-40** showed moderate potency toward all cell lines (average IC_50_ 12.7−20.7 µM). While alkaloid **34-32** displayed good antiproliferative activity (IC_50_ 8.7 µM) against KB cells, it was much less potent against A549, MDA-MB-231, and KB-VIN cells. Alkaloids **34-5**, **34-13**, **34-15**, **34-29**, **34-35**, **34-37**, and **34-41** exhibited only weak potency against all cell lines (average IC_50_ 22.0−26.5 µM). Finally, alkaloids **34-24**, **34-30**, and **34-34** were inactive against all five human tumor cell lines, while **34-12**, **34-33**, and **34-39** showed limited potency.

Among the derivatives esterified at both C-1 and -14, alkaloids **34-19** and **34-20** exhibited significant potency against all five tested cell lines (average IC_50_ 4.9 and 5.0 µM, respectively). Alkaloid **34-9** (average IC_50_ 11.9 µM) showed significant antiproliferative activity against MDA-MB- 231 and KB cells (IC_50_ 4.7 and 5.8 µM, respectively) comparable with **34-19** and **34-20**, but was less potent against MCF-7 and A549 (IC_50_ 12.2 and 24.8 µM, respectively) and inactive against KB-VIN. Alkaloid **34-23** exhibited only weak potency toward all cell lines (average IC_50_ 23.7 µM). Alkaloids **34-4**, **34-7**, **34-11**, **34-14**, **34-26**, **34-36**, **34-45**, **34-46**, and **34-47** were inactive against all five human tumor cell lines, while **34-22** and **34-28** showed limited potency.

Particularly, C-1 monoacylated delcosine derivatives (**34-3**, **34-6**, **34-8**, **34-10**, **34-13**, **34-21**, **34-25**, **34-27**, and **34-35**) were significantly more potent compared with corresponding C-1,14 diacylated delcosine derivatives (**34-4**, **34-7**, **34-9**, **34-11**, **34-14**, **34-22**, **34-23**, **34-26**, **34-28** and **34-36**). Thus, a C-1 acyloxy group and C-14 hydroxy group are crucial for enhanced antiproliferative activity of **1**-derivatives. Regarding alkaloids **34-18** (pentafluorobenzoate at C-1, hydroxy at C-14), **34-19** (pentafluorobenzoate at C-1 and C-14), and **34-20** (pentafluorobenzoate at C-1, acetate at C-14), all three alkaloids were essentially equipotent against three of the five tumor cell lines, while **34-18** was somewhat less potent than the diacylated alkaloids against MCF-7 and KB-VIN cells.

Striking observations from the data in Table 2 were the consistent identities of the most potent alkaloids. Alkaloids **34-8**, **34-10**, **34-19**, and **34-20** exhibited the highest potency against all five tested tumor cell lines with IC_50_ values ranging from 4.3 to 6.0 µM. The same range of potency was found with alkaloid **34-18** against A549 cells, with alkaloids **34-9** and **34-18** against MDA-MB-231 cells, and with **34-6**, **34-9**, and **34-18** against KB cells. The potencies of **34-6** and **34-17** (IC_50_ 5.6−11.8 µM) generally ranked somewhat below those of the most potent alkaloids, except against the MCF-7 cell line, where they were even less active.

The identity of the substituent(s) on the acyl group affected the antiproliferative potency. Notably, among the 1,14-diacyl and 1-acyl-14-acetyl derivatives, only alkaloids **34-19** and **34-20** with one or two pentafluorinated benzoyl esters, respectively, showed significant potency against all five tested cell lines. Alkaloid **34-9** with two 3-nitro-4-chlorobenzoyl groups showed good potency against certain cell lines. Similarly, the 1-monoacylated alkaloids with the highest potency against the five tumor cell lines contained 3-nitro-4-chloro- (**34-8**) and pentafluoro- (**34-18**) as well as 4-dichloro-methyl- (**34-10**) benzoyl esters. The chlorinated alkaloids **34-8** and **34-10** as well as **34-6**, which has 3,5-dichloro substitution on the benzoate ester, were more potent than **34-5** with only a single chloro group or **34-13** with chloro and fluoro groups. Similarly, alkaloid **34-18** showed increased antiproliferative activity against the five tumor cell lines compared with other fluorinated alkaloids **34-13**~**34-17**, **34-21**~**34-27**. Moreover, with some exceptions against certain cell lines, alkaloids with bromo (**34-3** and **34-4**), dimethylamino (**34-12**), dimethoxy (**34-29**), trimethoxy (**34-30**), diethoxy (**34-31**), benzyloxy (**34-32**), cyano (**34-33**), methylenedioxy (**34-34** and **34-35**), nitro (**34-45** and **34-46**), and ethoxy (**34-47**) substituted benzoate esters or phenylacetyl (**34-37**), cinnamoyl (**34-38** and **34-39**), 1-naphthoyl (**34-40**), and anthraquinone-2-carbonyl (**34-41**) esters were less potent or inactive. 

Interestingly, the active alkaloids were generally effective against P-gp overexpressing MDR subline KB-VIN, while alkaloids such as vincristine and paclitaxel are ineffective due to excretion from the MDR cells by P-gp. These results suggest that these diterpenoids are not substrates for P-gp.

### 2.3. Lycoctonine-Type (7,8-methylenedioxy) C_19_-Diterpenoid Alkaloids

The tested lycoctonine-type (7,8-methylenedioxy) C_19_-diterpenoid alkaloids included 19 natural alkaloids, delcorine (**36**), delpheline (**37**), pacinine (**38**), yunnadelphinine (**39**), melpheline (**40**), bonvalotidine C (**41**), *N*-deethyl-*N*-formylpaciline (**42**), *N*-deethyl-*N*-formylpacinine (**43**), isodel-pheline (**44**), pacidine (**45**), eladine (**46**), *N*-formyl-4,19-secopacinine (**47**), *N*-formyl-4,19-secoyunna-delphinine (**48**), iminoisodelpheline (**49**), iminodelpheline (**50**), laxicyminine (**51**), *N*-deethyl-19-oxo-isodelpheline (**52**), *N*-deethyl-19-oxodelpheline (**53**), and 19-oxoisodelpheline (**54**), purified from seeds of *Delphinium elatum* cv. Pacific Giant [35,39,40,41,42] (Figure 4). The remaining 22 tested C_19_-diterpenoids were synthetic derivatives (**37-1**~**37-22**) [43] prepared from **37** (Figure 4). 

All tested lycoctonine-type (7,8-methylenedioxy) C_19_-diterpenoid alkaloids were evaluated for antiproliferative activity against human tumor cell lines [30,40,41,42,43] (Table 3). The lycoctonine-type (7,8-methylenedioxy) C_19_-diterpenoid alkaloids, both the natural alkaloids (**36**~**54**) and synthetic analogs that did not contain a C-6 ester group (**37-20** and **37-22**), were inactive (IC_50_ > 20 or 40 μM). Among the C-6 esterified alkaloids, **37-1**, **37-17**, and **37-18** exhibited the highest average potency toward four tested cell lines (A549, DU145, KB and KB-VIN; average IC_50_ 9.83, 9.57, and 9.41 μM, respectively). Alkaloids **37-3**, **37-5**~**37-7**, **37-9**, **37-10**, **37-12**, **37-13**, **37-16**, and **37-19** showed moderate potency against all tested cell lines (average IC_50_ 13.9−20.8 µM). However, alkaloid **37-13** showed significantly increased cytotoxic activity (IC_50_ 10.2 μM) against A549 cells compared with **37-1**, **37-17**, and **37-18**, but was generally less potent against DU145 and KB cells. While alkaloids **37-12**, **37-13**, **37-16**, and **37-19** displayed good antiproliferative activity (IC_50_ 6.8, 9.1, 6.5, and 4.7 µM, respectively) against KB-VIN cells, they were much less potent against A549, DU145, and KB cells. Alkaloids **37-4** and **37-21** were inactive against all tested cancer cell lines, while **37-2**, **37-8**, **37-11**, and **37-14** exhibited only weak potency toward all cell lines (average IC_50_ 23.0−29.2 µM).

The most noticeable observations from the data in Table 3 were the degree and relative ratio of KB/KB-VIN potency. Among the four cancer cell lines tested, the highest potency was found against the KB-VIN cell line by alkaloids **37-17**~**37-19** (IC_50_ 4.22, 4.40, and 4.71 μM, respectively), followed by alkaloids **37-16**, **37-12**, **37-1**, **37-13**, and **37-9** (IC_50_ 6.50, 6.80, 8.27, 9.10, and 11.9 μM, respectively). Generally, all active alkaloids showed the highest potency against the KB-VIN cell line compared with the other three tested cancer cell lines. Moreover, alkaloids **37-12**, **37-16**, **37-13**, and **37-19** showed over two-fold selectivity between the two cell lines (ratio of IC_50_ KB/IC_50_ KB-VIN: 2.15, 2.28, 2.31, and 2.57, respectively). Alkaloids **37-2**, **37-5**, and **37-17** displayed weak selectivity between the KB and KB-VIN cell lines (ratio of IC_50_ KB/IC_50_ KB-VIN: 1.55, 1.36, and 1.62, respectively). Finally, alkaloids **37-1**, **37-3**, **37-6**~**37-9**, **37-11**, **37-14**, **37-15**, and **37-18** displayed similar potency against the KB and KB-VIN cell lines (ratio of IC_50_ KB/IC_50_ KB-VIN: 1.07, 1.17, 1.06, 1.21, 1.04, 1.25, 1.07, 1.07, 1.17, and 1.23, respectively).

The identity of the substituent on the C-6 acyl group affected the cytotoxic potency. For instance, the alkaloids with the highest potency against the KB-VIN cell line contained chloro (**37-1**), fluoro (**37-12**, **37-18**, and **37-19**), trifluoromethyl (**37-9**, **37-13**, and **37-18**), ethoxy (**37-16**), or benzyloxy (**37-17**) substituents on the acyl group. Against the KB-VIN cell line, alkaloids **37-18** and **37-19** with both fluoro and trifluoromethyl/methyl groups were more potent than **37-9** with only a single trifluoromethyl group and even more potent than **37-2** with a single fluoro group. Similarly, alkaloid **37-13** showed increased cytotoxic activity against most cell lines compared with the related fluorinated alkaloids **37-14** and **37-15**. Moreover, alkaloids with nitro, methoxy, phenyl, trifluoromethoxy, trifluoromethythio, and methyl carboxylate groups on a C-6 benzoate ester were generally less potent.

## 3. Antiproliferative Activity of C_20_-Diterpenoid Alkaloid Derivatives

### 3.1. Actaline and Napelline-Type C_20_-Diterpenoid Alkaloids

One natural actaline-type C_20_-diterpenoid alkaloid [44], aconicarchamine A (**55**), isolated from rhizoma of *Aconitum japonicum* subsp. *subcuneatum* [30], (Figure 5) and seven natural napelline-type C_20_-diterpenoid alkaloids, lucidusculine (**57**), flavadine (**58**), 12-acetyllucidusculine (**59**), 1-acetyl-luciculine (**60**), dehydrolucidusculine (**61**), dehydroluciculine (**62**), and 12-acetyldehydroluciduscu-line (**63**), purified from roots of *Aconitum yesoense* var. *macroyesoense* [31,32,33], (Figure 5) were tested. Seven synthetic napelline-type C_20_-diterpenoid alkaloid derivatives (**56-1**~**56-7**) [18,32,45] were prepared from luciculine (**56**) (Figure 5) and tested also. All tested actaline- and napelline-type C_20_-diterpenoid alkaloids were evaluated for antiproliferative activity against four to five human tumor cell lines [28,30] (Table 4). Tested actaline- and napelline-type C_20_-diterpenoid alkaloids, both the natural alkaloids (**55** and **57**~**63**) and synthetic analogs (**56-1**~**56-4**, **56-6**, and **56-7**), were inactive (IC_50_ > 20 or 40 μM). Among the synthetic alkaloids, alkaloid **56-5** exhibited only weak potency toward the tested cell lines (A549, DU145, KB and KB-VIN; average IC_50_ 27.8 μM). Because the related alkaloids **57**, **60, 56-2**~**56-4**, **56-6**, and **56-7** were inactive against all tested cancer cell lines, a C-1 hydroxy group, C-12 acyloxy group, and C-15 acetoxy group found in **56-5** could be needed for antiproliferative activity of luciculine derivatives.

### 3.2. Hetisine-Type (Analogs of Kobusine) C_20_-Diterpenoid Alkaloids 

Tested hetisine-type (analogs of kobusine) C_20_-diterpenoid alkaloids included four natural alkaloids, ryosenamine (**64**), 9-hydroxynominine (**65**), and torokonine (**66**), isolated from rhizoma of *Aconitum japonicum* subsp. *subcuneatum* [27,28] (Figure 6) and kobusine (**67**), purified from roots of *Aconitum yesoense* var. *macroyesoense* [31] (Figure 6). Nineteen synthetic derivatives (**67-1**~**67-19**) [18,21,30,46,47] (Figure 6) prepared from **67** were tested also.

All tested hetisine-type (kobusine analogs) C_20_-diterpenoid alkaloids were evaluated for antiproliferative activity against four human tumor cell lines [27,28,30] (Table 5). Fifteen of the 23 alkaloids, both natural (**64**~**67**) and synthetic (**67-1**~**67-4**, **67-6**, **67-9**, **67-11**, **67-12**, **67-15**~**67-17**), were inactive (IC_50_ > 20 or 40 μM). Kobusine derivatives **67-5**, **67-7**, **67-10**, **67-18**, and **67-19** exhibited the highest average potency over the four tested cell lines (A549, DU145, KB and KB-VIN; average IC_50_ 7.8, 6.1, 6.2, 6.8, and 4.7 μM, respectively), and alkaloids **67-8**, **67-13**, and **67-14** showed moderate potency (average IC_50_ 16.6, 14.3, and 11.6 µM, respectively). However, while alkaloid **67-14** showed good cytotoxic activity (IC_50_ 9.6 μM) against DU145 cells, it was much less potent against A549, KB, and KB-VIN cells. 

Among these analogs of **67**, esterification of C-15 in addition to C-11 increased potency significantly (compare **67-8** to **67-10**) or even converted an inactive to an active alkaloid (compare **67-3** to **67-5**, **67-6** to **67-7**, **67-16** to **67-18**). Consequently, all of the most potent analogs (**67-5**, **67-7**, **67-10**, **67-18**, and **67-19**) of **67** were esterified at both C-11 and C-15. 

Striking observations from the data in Table 5 were the degree and comparative ratio of KB/KB-VIN potency. Five alkaloids (**67-5**, **67-7**, **67-10**, **67-18**, and **67-19**) were quite potent (IC_50_ < 10 µM) against KB-VIN. Indeed, alkaloid **67-19** exhibited a significantly low IC_50_ value of 3.1 µM. The ratios of KB to KB-VIN (IC_50_ KB/IC_50_ KB-VIN) were greater than 0.73 for all active alkaloids, with many alkaloids displaying comparable potency against the two cell lines, in contrast with paclitaxel (ratio of 0.0067). Alkaloid **67-19** showed over 1.3-fold selectivity with the greatest cytotoxic activity against KB-VIN (IC_50_ KB/IC_50_ KB-VIN: 1.32).

### 3.3. Hetisine-Type (Analogs of Pseudokobusine) C_20_-Diterpenoid Alkaloids 

The two tested natural hetisine-type (analogs of pseudokobusine) C_20_-diterpenoid alkaloids pseudokobusine (**68**) and 15-veratroylpseudokobusine (**68-11**) were purified from the roots of *Aconitum yesoense* var. *macroyesoense* [31,32] (Figure 7). The 36 tested synthetic derivatives (**68-1**~**68-10**, **68-12**~**68-37**) [18,21,30,32,46,47,48,49] (Figure 7) were prepared from **68**.

All tested hetisine-type (**68** analogs) C_20_-diterpenoid alkaloids were evaluated for antiproliferative activity against four human tumor cell lines [30] (Table 6). Many alkaloids, both natural alkaloids (**68** and **68-11**) and synthetic analogs (**68-1**~**68-3**, **68-6**, **68-8**, **68-9**, **68-14**, **68-16**~**68-18**, **68-21**, **68-23**, **68-25**~**68-31**, **68-33**~**68-37**), were inactive (IC_50_ > 20 µM). The pseudokobusine derivatives **68-5**, **68-15**, **68-19**, **68-20**, **68-24**, and **68-32** exhibited the highest average potency over the tested cell lines (A549, DU145, KB and KB-VIN; average IC_50_ 7.0, 5.2, 5.3, 7.4, 7.1, and 6.1 µM, respectively). Alkaloids **68-7**, **68-10**, **68-12**, **68-13**, and **68-22** showed moderate potency over all tested cell lines (average IC_50_ 13.5-16.8 µM). However, although alkaloid **68-10** showed good cytotoxic activity (IC_50_ 8.0 µM) against A549 cells, it was much less potent against DU145, KB, and KB-VIN cells.

Among the analogs of **68**, four C-11 mono-substituted alkaloids (**68-15**, **68-20**, **68-24**, and **68-32**) and two C-11,15 di-esterified alkaloids (**68-5** and **68-19**) exhibited average IC_50_ values of less than 10 μM. Certain C-11 (**68-7**, **68-10**, and **68-22**), C-6,11 (**68-4** and **68-12**) and C-6,15 (**68-13**) esterified alkaloids were generally less potent, while all C-6 (**68-3**, **68-6**, **68-14**, and **68-23**) and C-15 (**68-1**, **68-11**, **68-16**, **68-25**, **68-28**, and **68-30**) mono-substituted alkaloids, as well as the tri-substituted analog (**68-18**), were inactive. Thus, all more active (IC_50_ < 10 µM) C_20_-diterpenoid alkaloids in this classification had an ester or ether group on the C-11 hydroxy and were 11-monoester/11,15-diester analogs of **68** (OH at C-6).

The data in Table 6 led to noticeable observations about the degree and comparative ratio of KB/KB-VIN potency. Six alkaloids (**68-5**, **68-15**, **68-19**, **68-20**, **68-24**, and **68-32**) were quite potent (IC_50_ < 10 µM) against KB-VIN. Indeed, alkaloid **68-32** exhibited a low IC_50_ value of 5.2 µM. The ratios of KB to KB-VIN (IC_50_ KB/IC_50_ KB-VIN) were greater than 0.70 for all active alkaloids, with many alkaloids displaying comparable potency against the two cell lines, in contrast with paclitaxel (ratio of 0.0067). Alkaloids **68-12**, **68-13**, and **68-20** showed over 1.3-fold selectivity with their greatest cytotoxic activity against KB-VIN (IC_50_ KB/IC_50_ KB-VIN: 1.34, 1.48, and 1.44, respectively).

In mechanism of action studies on selected diterpenoid alkaloids, the hetisine-type C_20_-diterpenoid alkaloid derivatives **68-7** and **68-22** showed important suppressive effects against Raji cells. Further study indicated that **68-22** inhibited extracellular signal-regulated kinase phosphorylation but induced enhanced phosphoinositide 3 kinase phosphorylation, leading to accumulation of Raji cells in the G1 or sub G1 phase [20]. More investigation is certainly warranted.

## 4. Discussion

We have synthesized acylated derivatives of various C_19_- and C_20_-diterpenoid alkaloids. Totally, 199 natural alkaloids and their derivatives were evaluated against four to five human tumor cell lines. Among all alkaloids, 128 alkaloids were non-toxic (IC_50_ > 20 or 40 µM) and 51 alkaloids showed moderate antiproliferative effects (average IC_50_ = 10–40 µM). General summaries are described briefly below, and the most active compounds are shown in Figure 8.

Among the aconitine-type C_19_-diterpenoid alkaloids, the fatty acid ester at C-8 and the anisoyl group at C-14 found in **17** may be important to the cytotoxic activity. Compounds without the fatty acid ester at C-8 were inactive, and compounds with an unsubstituted benzoyl group at C-14 were less potent.

Among the C_19_-diterpenoid alkaloids, the most active alkaloids were lycoctonine-type C_19_-diterpenoid alkaloids with two different substitution patterns, C-1 (delcosine derivatives) and C-6 (delpheline derivatives). Delcosine derivatives **34-6**, **34-8**, **34-10**, and **34-18**, which are acylated at the C-1 hydroxy, as well as delpheline derivatives **37-1**, **37-17**, and **37-18**, which are acylated at the C-6 hydroxy, exhibited the greatest potency over all tested cell lines, including MDR KB-VIN. 

Among the lycoctonine-type (7,8-diol) C_19_-diterpenoid alkaloids, a C-1 acyloxy group and C-14 hydroxy group were important for improved antiproliferative activity. The C-1,14 diacylated delcosine derivatives were generally less potent than corresponding C-1 monoacylated delcosine derivatives. The 1-monoacylated alkaloids with the highest potency (IC_50_ 4−6 µM) against five tested cell lines contained 3-nitro-4-chloro- (**34-8**) and pentafluoro- (**34-18**) as well as 4-dichloromethyl- (**34-10**) benzoyl esters. Two or one pentafluorinated benzoyl esters were also found in the two most consistently potent alkaloids (**34-19** and **34-20**) among the 1,14-diacyl and 1-acyl-14-acetyl derivatives.

Among the lycoctonine-type (7,8-methylenedioxy) C_19_-diterpenoid alkaloids, none of the tested compound reached the potency levels of the most active 7,8-diol compounds. However, three 6-acylated delpheline derivatives **37-17**~**37-19** did show significant potency against the KB-VIN cell line (IC_50_ 4.22, 4.40, and 4.71 μM, respectively). Interestingly, the two latter compounds contained fluorinated benzoyl esters. In addition, among 19 tested delpheline derivatives, four compounds (**37-12**, **37-16**, **37-13**, and **37-19**) showed over two-fold selectivity between the MDR and parental cell lines (ratio of IC_50_ KB/IC_50_ KB-VIN: 2.15, 2.28, 2.31, and 2.57, respectively). 

None of the 15 tested actaline- and napelline-type C_20_-diterpenoid alkaloids showed significant antiproliferative potency. Only 12-benzoyllucidsuculine (**56-5**) with C-1 hydroxy, C-12 acyloxy, and C-15 acetoxy groups showed even weak potency.

Among C_20_-diterpenoid alkaloids, the most active alkaloids were hetisine-type C_20_-diterpenoid alkaloids with two different substitution patterns, C-11,15 (kobusine) and C-6,11,15 (pseudo-kobusine). Hetisine-type C_20_-diterpenoid alkaloids **67-5**, **67-7**, **67-10**, **67-18**, **67-19**, **68-5**, **68-15**, **68-19**, **68-20**, **68-25**, and **68-32**, which are acylated or tritylated at the C-11 hydroxyl, exhibited the greatest potency over all tested cell lines, including MDR KB-VIN. All five most active kobusine derivatives (**67-5**, **67-7**, **67-10**, **67-18**, and **67-19**) are acylated at both C-11 and C-15. All tested derivatives with a hydroxy group at either C-11 or C-15 were inactive or much less active. All six most active pseudo-kobusine derivatives (**68-5**, **68-15**, **68-19**, **68-20**, **68-25**, and **68-32**) contain a free hydroxy group at C-6. The substituent at C-11 is either a benzoyl/cinnamoyl ester (**68-5**, **68-15**, **68-19**, **68-20**, and **68-25**) or a trityl ether (**68-32**). Finally, the moiety at C-15 is a hydroxy group (**68-15**, **68-20**, **68-25**, and **68-32**) or benzoyl ester (**68-5**, **68-19**). 

Furthermore, previously our study, Antitumor properties and radiation-sensitizing effects of various types of novel derivatives prepared from C_19_- and C_20_-diterpenoid alkaloids were also investigated [19]. Two novel hetisine-type C_20_-diterpenoid derivatives (**68-7** and **68-20**) showed significant suppressive effects against the Raji non-Hodgkin’s lymphoma cell line [20].

## 5. Conclusions

We have synthesized acylated derivatives of various C_19_- and C_20_-diterpenoid alkaloids. All alkaloids and their derivatives were screened against four to five human tumor cell lines. Alkaloids **37-2**, **37-9**, **37-17**, **37-18**, **56-5**, **67-7**, **67-14**, **67-19**, **68-4**, **68-12**, **68-20**, **68-22**, **68-24**, and **68-32** showed comparable potency against KB and KB-VIN cancer cell lines, and some alkaloids showed tumor- selective activity. Alkaloids **37-12**, **37-13**, **37-16**, and **37-19** exhibited greater inhibitory activity against drug-resistant KB-VIN cells (2.15~2.57-fold) than the parental KB cells. These results demonstrate that modified lycoctonine-type C_19_-diterpenoid alkaloids and hetisine-type C_20_-diterpenoid alkaloids are not substrates of P-gp and could be effective against P-gp overexpressing MDR tumors. These promising new lead alkaloids merit continued studies to evaluate their potential as antitumor agents, particularly with enhanced resistant tumor selectivity. In addition, our results from modification-based antitumor activity studies can be used for further development of anticancer drugs overcoming an MDR phenotype.

## Figures and Tables

**Figure 1 molecules-24-02317-f001:**
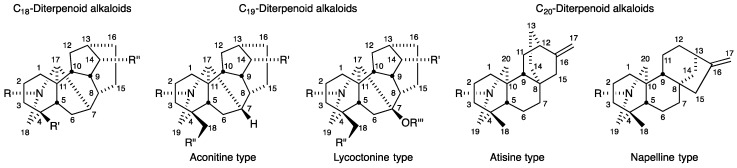
Classifications, general structures and numbering systems for C_18_-, C_19_-, and C_20_-diterpenoid alkaloids.

**Figure 2 molecules-24-02317-f002:**
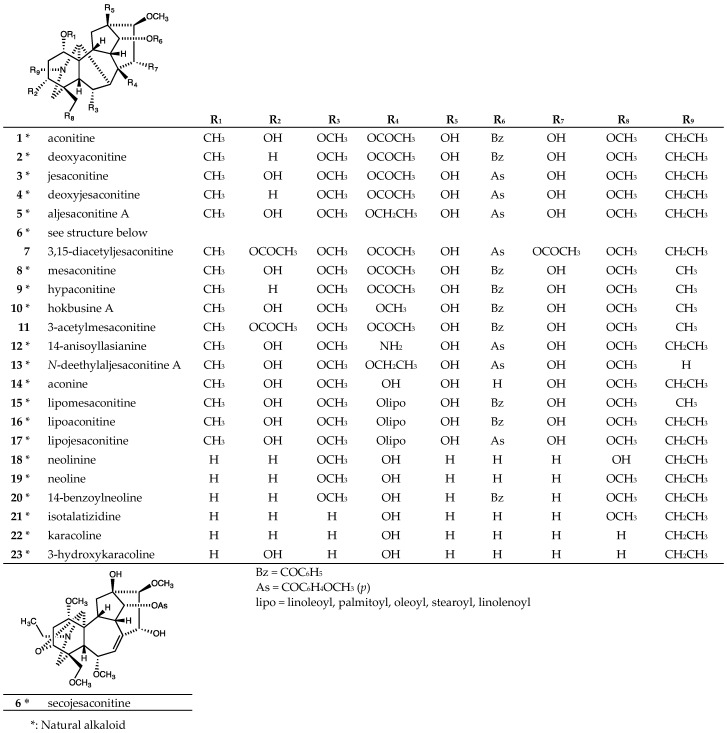
Chemical structures of aconitine-type C_19_-diterpenoid alkaloids **1**–**23**.

**Figure 3 molecules-24-02317-f003:**
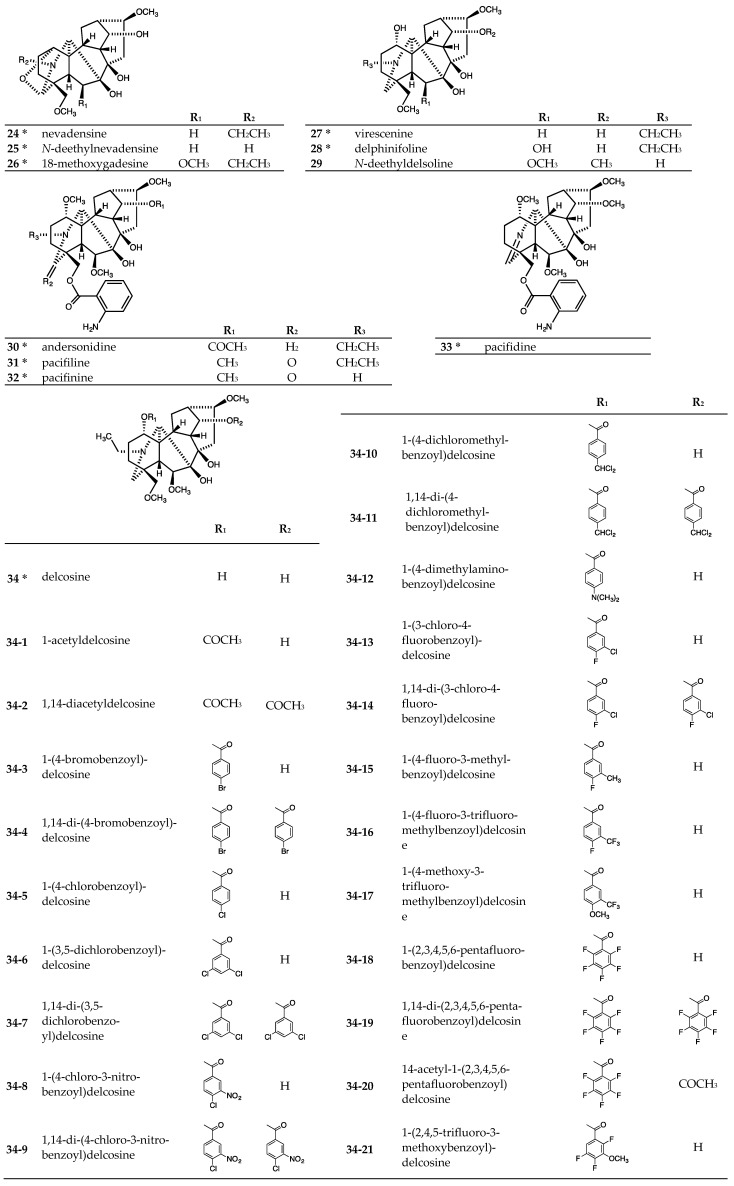
Chemical structures of lycoctonine-type (7,8-diol) C_19_-diterpenoid alkaloids **24**–**35**.

**Figure 4 molecules-24-02317-f004:**
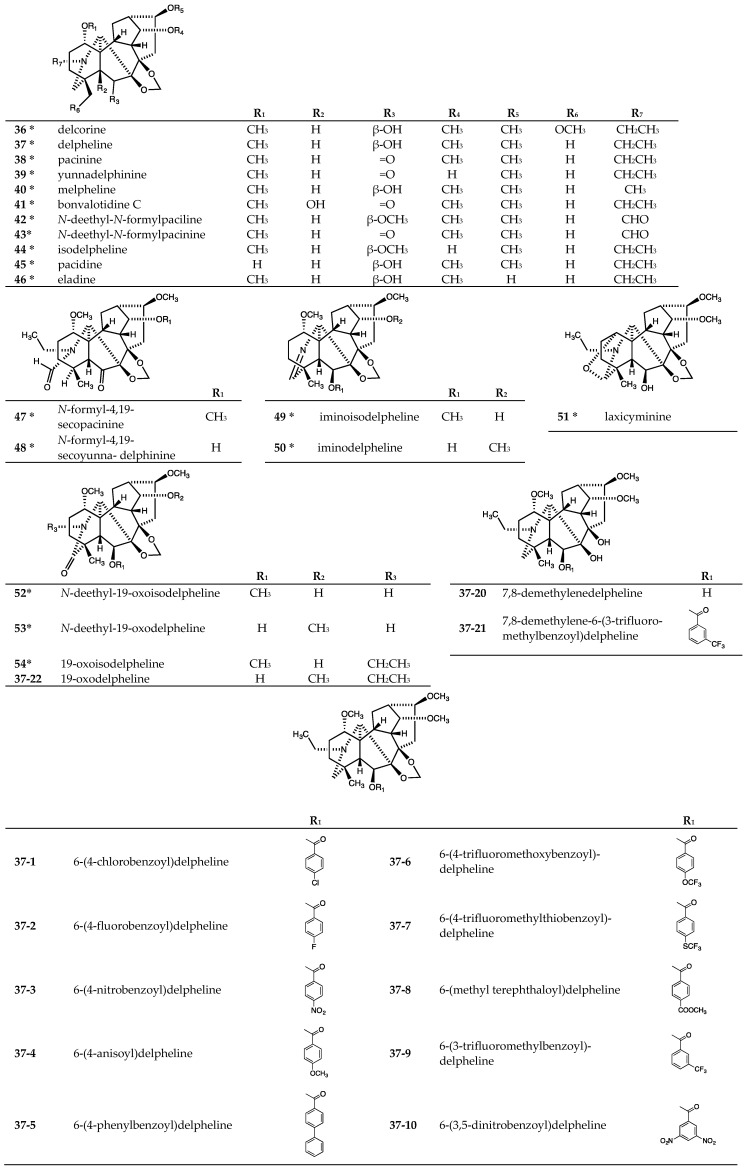
Chemical structures of lycoctonine-type (7,8-methylenedioxy) C_19_-diterpenoid alkaloids **36**-**54**.

**Figure 5 molecules-24-02317-f005:**
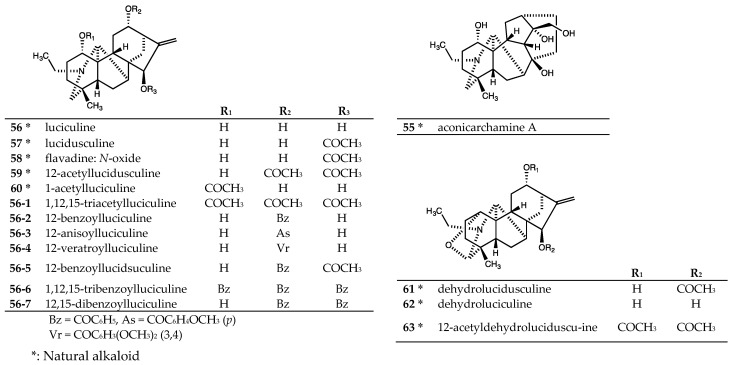
Chemical structures of actaline and napelline-type C_20_-diterpenoid alkaloids **55**~**63**.

**Figure 6 molecules-24-02317-f006:**
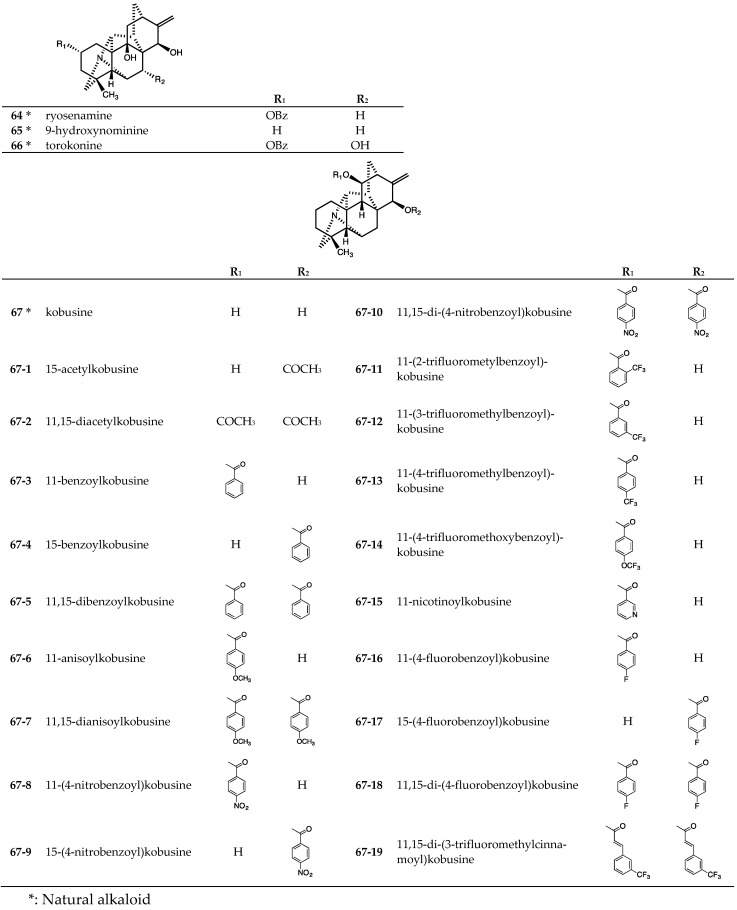
Chemical structures of hetisine-type (analogs of kobusine) C_20_-diterpenoid alkaloids **64**~**67-19**.

**Figure 7 molecules-24-02317-f007:**
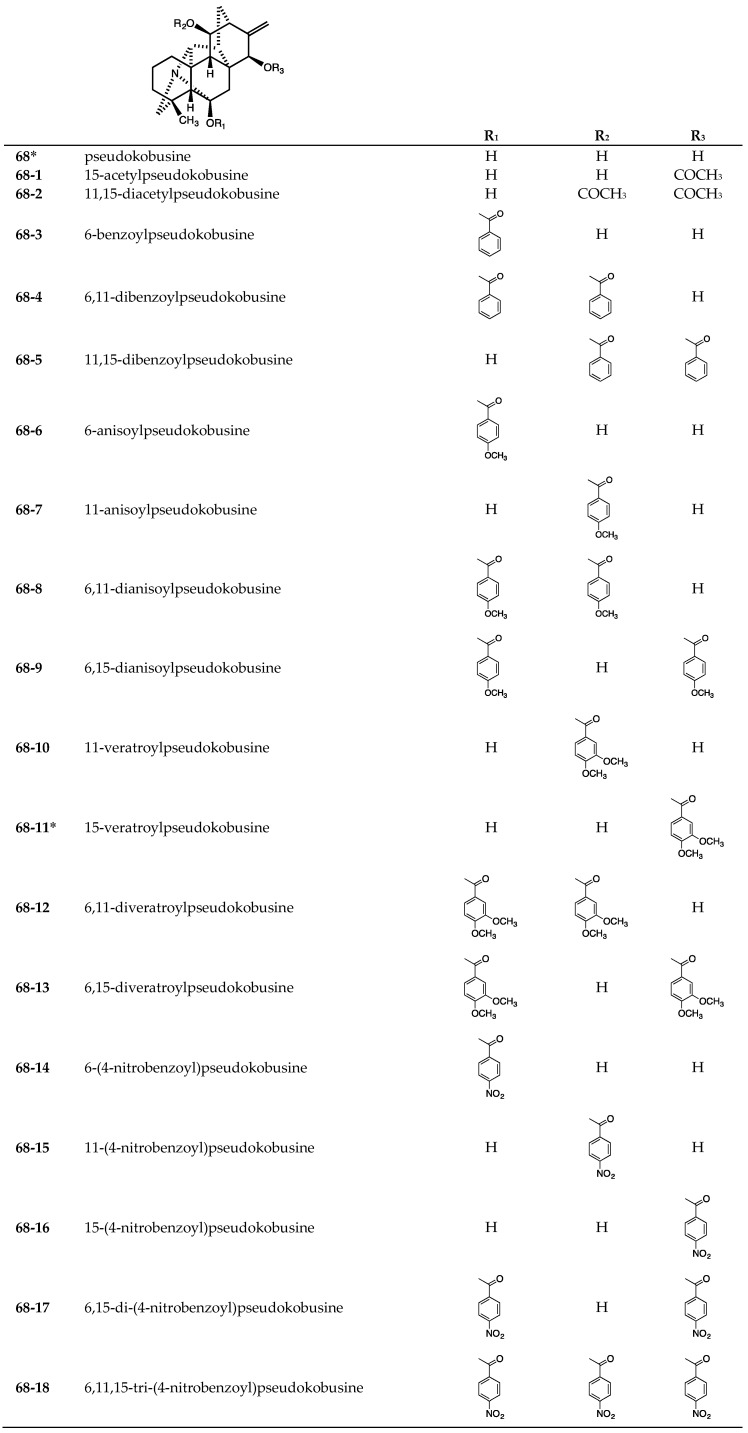
Chemical structures of hetisine-type (analogs of pseudokobusine) C_20_-diterpenoid alkaloids **68**~**68-37**.

**Figure 8 molecules-24-02317-f008:**
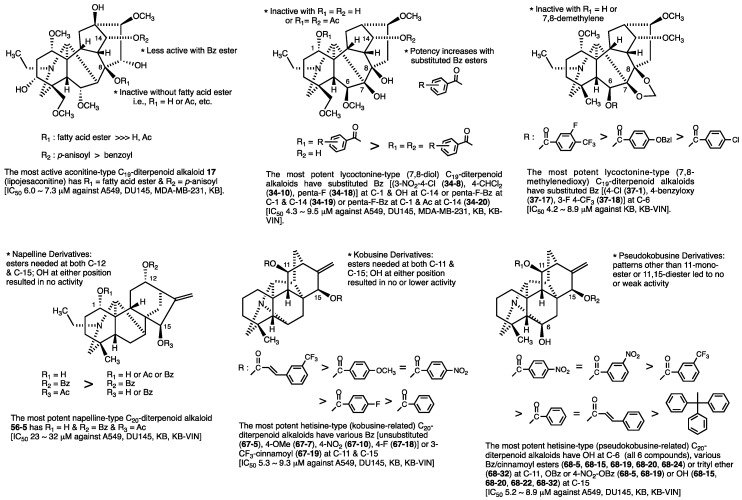
Most potent tested diterpenoid alkaloids & structure-activity correlations.

**Table 1 molecules-24-02317-t001:** Cytotoxic activity data for aconitine-type C_19_-diterpenoid alkaloids and derivatives **1**–**23**.

	Cell Line/IC_50_ (μM) ^1^					
Alkaloids	A549	DU145	MDA-MB-231	MCF-7	KB	KB-VIN
Aconitine (**1**)	>20	>20	-	-	>20	>20
Deoxyaconitine (**2**)	>20	>20	-	-	>20	>20
Jesaconitine (**3**)	>20	>20	-	-	>20	>20
Deoxyjesaconitine (**4**)	>20	>20	-	-	>20	>20
Aljesaconitine A (**5**)	>20	>20	-	-	>20	>20
Secojesaconitine (**6**)	>20	>20	-	-	>20	>20
**7**	>20	>20	-	-	>20	>20
Mesaconitine (**8**)	>20	>20	-	-	>20	>20
Hypaconitine (**9**)	>20	>20	-	-	>20	>20
Hokbusine A (**10**)	>20	>20	-	-	>20	>20
**11**	>20	>20	-	-	>20	>20
14-Anisoyllasianine (**12**)	>40	-	>40	>40	>40	>40
*N*-Deethylaljesaconitine A (**13**)	>40	-	>40	>40	>40	>40
Aconine (**14**)	>40	-	>40	>40	>40	>40
Lipomesaconitine (**15**)	17.2 ± 2.3	-	20.0 ± 0.2	19.0 ± 1.0	10.0 ± 3.3	21.5 ± 0.9
Lipoaconitine (**16**)	17.4 ± 1.1	-	15.5 ± 0.5	16.0 ± 0.3	13.7 ± 1.3	20.3 ± 1.1
Lipojesaconitine (**17**)	7.3 ± 0.3	-	6.0 ± 0.2	6.7 ± 0.2	6.0 ± 0.2	18.6 ± 0.9
Neolinine (**18**)	>40	-	>40	>40	>40	>40
Neoline (**19**)	>20	>20	-	-	>20	>20
14-benzoylneoline (**20**)	>20	>20	-	-	>20	>20
Isotalatizidine (**21**)	>40	-	>40	>40	>40	>40
Karacoline (**22**)	>20	>20	-	-	>20	>20
3-Hydroxykaracoline (**23**)	>40	-	>40	>40	>40	>40
**PXL**^2^ (nM)	4.8 ± 0.6	5.9 ± 1.9	8.4 ± 0.8	10.2 ± 0.9	5.8 ± 0.2	2405.4 ± 44.8

^1^ Values are means ± standard deviation; ^2^ Paclitaxel (PXL; nM) was used as an experimental control.

**Table 2 molecules-24-02317-t002:** Cytotoxic activity data for lycoctonine-type (7,8-diol) C_19_-diterpenoid alkaloids and synthetic analogs of delcosine **24**~**35**.

	Cell Line/IC_50_ (μM) ^1^					
Alkaloids	A549	DU145	MDA-MB-231	MCF-7	KB	KB-VIN
Nevadensine (**24**)	>40	-	>40	>40	>40	>40
*N*-Deethylnevadensine (**25**)	>40	-	>40	>40	>40	>40
18-Methoxygadesine (**26**)	>20	>20	-	-	>20	>20
Virescenine (**27**)	>40	-	>40	>40	>40	>40
Delphinifoline (**28**)	>20	>20	-	-	>20	>20
*N*-Deethyldelsoline (**29**)	>20	>20	-	-	>20	>20
Andersonidine (**30**)	>20	>20	-	-	>20	>20
Pacifiline (**31**)	>20	>20	-	-	>20	>20
Pacifinine (**32**)	>20	>20	-	-	>20	>20
Pacifidine (**33**)	>20	>20	-	-	>20	>20
Delcosine (**34**)	>20	>20	-	-	>20	>20
**34-1**	>20	>20	-	-	>20	>20
**34-2**	>20	>20	-	-	>20	>20
**34-3**	20.6 ± 0.3	-	19.4± 1.0	17.9 ± 0.3	14.6 ± 0.6	17.1 ± 0.8
**34-4**	>40	-	>40	>40	>40	>40
**34-5**	18.7 ± 0.1	-	29.1 ± 1.6	25.8 ± 1.4	19.6 ± 0.3	21.1 ± 1.5
**34-6**	7.7 ± 0.9	-	8.6 ± 6.0	15.8 ± 4.2	5.6 ± 1.2	8.6 ± 1.9
**34-7**	>40	-	>40	>40	>40	>40
**34-8**	4.5 ± 0.5	-	5.0 ± 0.1	5.9 ± 0.3	5.4 ± 0.3	5.6 ± 0.4
**34-9**	24.8 ± 0.1	-	4.7 ± 0.1	12.2 ± 0.3	5.8 ± 0.4	>40
**34-10**	4.8 ± 0.3	-	4.8 ± 0.7	5.7 ± 0.4	4.3 ± 0.5	5.3 ± 0.4
**34-11**	>40	-	>40	>40	>40	>40
**34-12**	26.5 ± 0.3	-	>40	40.6 ± 2.5	27.8 ± 1.7	28.1 ± 3.0
**34-13**	20.8 ± 1.7	-	32.4 ± 1.8	25.9 ± 2.4	23.0 ± 2.4	21.5 ± 1.3
**34-14**	>40	-	>40	>40	>40	>40
**34-15**	21.7 ± 1.6	-	30.2 ± 2.7	26.9 ± 1.4	20.7 ± 1.2	21.5 ± 3.6
**34-16**	14.4 ± 2.1	-	20.1 ± 0.7	16.4 ± 2.1	13.6 ± 1.1	15.7 ± 0.8
**34-17**	11.4 ± 1.4	-	10.4 ± 1.7	22.5 ± 1.5	10. 8 ± 1.9	11.8 ± 3.2
**34-18**	4.7 ± 0.1	-	5.3 ± 0.2	9.2 ± 0.4	5.8 ± 0.6	9.5 ± 0.5
**34-19**	4.9 ± 0.1	-	4.9 ± 0.1	5.3 ± 0.3	4.7 ± 0.1	4.9 ± 0.1
**34-20**	4.8 ± 0.1	-	4.6 ± 0.3	6.0 ± 0.1	4.8 ± 0.4	4.9 ± 0.4
**34-21**	20.8 ± 2.1	-	21.5 ± 0.6	21.4± 0.3	18.6 ± 1.7	15.0 ± 0.1
**34-22**	>40	-	>40	>40	>40	39.1 ± 2.0
**34-23**	23.8 ± 2.0	-	25.2 ± 1.0	23.3 ± 1.1	23.7 ± 1.1	22. 6 ± 0.3
**34-24**	>20	>20	-	-	>20	>20
**34-25**	20.6 ± 1.2	-	21.3 ± 1.3	22.4 ± 1.2	20. 8 ± 2.1	18.0 ± 1.0
**34-26**	>40	-	>40	>40	>40	>40
**34-27**	18.6 ± 2.6	-	19.7 ± 2.0	20.6 ± 1.2	22.2± 1.8	19.8 ± 1.9
**34-28**	33.0 ± 2.1	-	32.4 ± 1.7	31.1 ± 0.8	23.2 ± 1.1	40.0 ± 1.0
**34-29**	23.8 ± 2.6	-	33.4 ± 1.7	29.8 ± 1.2	22.8 ± 1.7	22.6 ± 2.4
**34-30**	>40	-	>40	>40	>40	>40
**34-31**	17. 3 ± 2.2	-	23. 1 ± 0.5	20.0 ± 0.7	16.2 ± 1.8	17.4 ± 1.9
**34-32**	16.5 ± 1.3	-	22.5 ± 0.8	-	8.71 ± 0.7	15.8 ± 0.8
**34-33**	40.9 ± 5.3	-	>40	>40	36.3 ± 1.0	29.3 ± 0.6
**34-34**	>40	-	>40	>40	>40	>40
**34-35**	21.2 ± 0.1	-	24.8 ± 1.6	24.6 ± 1.0	18.7 ± 1.2	21.7± 0.6
**34-36**	>40	-	>40	>40	>40	>40
**34-37**	23.8 ± 0.5	-	32.9 ± 1.0	22.6 ± 1.5	21.2 ± 0.1	19.2 ± 0.1
**34-38**	11.2 ± 0.7	>20	-	-	21.1 ± 3.9	19.5 ± 8.2
**34-39**	29.7 ± 0.7	-	43.2 ± 1.8	32.0 ± 0.6	36.0 ± 0.4	45.1 ± 3.4
**34-40**	18.5 ± 0.5	-	17.9 ± 0.5	15.5± 0.6	13.7± 0.1	14.2 ± 0.5
**34-41**	22.9± 0.5	-	20.7 ± 2.1	20.5 ± 1.0	21.6 ± 0.1	24.4 ± 0.5
**34-42**	>20	>20	-	-	>20	>20
14-Acetyldelcosine (**34-43**)	>20	>20	-	-	>20	>20
**34-44**	>20	>20	-	-	>20	>20
**34-45**	>20	>20	-	-	>20	>20
**34-46**	>20	>20	-	-	>20	>20
**34-47**	>20	>20	-	>20	>20	>20
14-Acetylbrowniine (**35**)	>20	>20	-	-	>20	>20
**PXL**^2^ (nM)	4.8 ± 0.6	5.9 ± 1.9	8.4 ± 0.8	10.2 ± 0.9	5.8 ± 0.2	2405.4 ± 44.8

^1^ Values are means ± standard deviation; ^2^ Paclitaxel (PXL; nM) was used as an experimental control.

**Table 3 molecules-24-02317-t003:** Cytotoxic activity data for lycoctonine-type (7,8-methylenedioxy) C_19_-diterpenoid alkaloids and synthetic analogs of delpheline **36**~**54**.

	Cell Line/IC_50_ (μM) ^1^					KB/KB-VIN Ratio
Alkaloids	A549	DU145	MDA-MB-231	KB	KB-VIN	
Delcorine (**36**)	>40	-	>40	>40	>40	-
Delpheline (**37**)	>20	>20	-	>20	>20	-
Pacinine (**38**)	>20	>20	-	>20	>20	-
Yunnadelphinine (**39**)	>20	>20	-	>20	>20	-
Melpheline (**40**)	>40	-	>40	>40	>40	-
Bonvalotidine C (**41**)	>40	-	>40	>40	>40	-
*N*-Deethyl-*N*-formylpaciline (**42**)	>40	-	>40	>40	>40	-
*N*-Deethyl-*N*-formylpacinine (**43**)	>40	-	>40	>40	>40	-
Isodelpheline (**44**)	>40	-	>40	>40	>40	-
Pacidine (**45**)	>40	-	>40	>40	>40	-
Eladine (**46**)	>40	-	>40	>40	>40	-
*N*-Formyl-4,19-secopacinine (**47**)	>40	-	>40	>40	>40	-
*N*-Formyl-4,19-secoyunnadelphinine (**48**)	>40	-	>40	>40	>40	-
Iminoisodelpheline (**49**)	>40	-	>40	>40	>40	-
Iminodelpheline (**50**)	>40	-	>40	>40	>40	-
Laxicyminine (**51**)	>40	-	>40	>40	>40	-
*N*-Deethyl-19-oxoisodelpheline (**52**)	>40	-	>40	>40	>40	-
*N*-Deethyl-19-oxo-delpheline (**53**)	>40	-	>40	>40	>40	-
19-Oxoisodelpheline (**54**)	>40	-	>40	>40	>40	-
**37-1**	14.8 ± 3.8	7.4 ± 1.2	-	8.9 ± 2.0	8.3 ± 1.6	1.07
**37-2**	38.1 ± 11.8	15.6 ± 5.4	-	23.3 ± 3.9	15.0 ± 6.5	1.55
**37-3**	22.7 ± 0.3	17.2 ± 3.3	-	20.7 ± 0.9	17.7 ± 3.5	1.17
**37-4**	>20	>20	-	>20	>20	-
**37-5**	24.1 ± 2.7	17.1 ± 11.4	-	23.6 ± 0.4	17.4 ± 7.4	1.36
**37-6**	18.7 ± 6.6	20.3 ± 7.1	-	20.1 ± 7.6	18.9 ± 5.0	1.06
**37-7**	21.1 ± 9.2	16.6 ± 12.7	-	21.7 ± 11.6	17.9 ± 4.2	1.21
**37-8**	28.7 ± 13.6	28.7 ± 7.2	-	24.3 ± 5.7	23.3 ± 3.7	1.04
**37-9**	21.2 ± 4.7	12.6 ± 3.0	-	14.9 ± 4.9	11.9 ± 3.3	1.25
**37-10**	20.9 ± 4.3	22.7 ± 6.0	-	19.1 ± 4.8	20.3 ± 2.7	0.94
**37-11**	30.8 ± 13.3	28.9 ± 4.7	-	29.5 ± 3.5	27.5 ± 3.1	1.07
**37-12**	19.9 ± 10.1	16.9 ± 6.7	-	14.6 ± 7.1	6.80 ± 5.0	2.15
**37-13**	10.2 ± 2.6	15.1 ± 6.0	-	21.0 ± 9.4	9.10 ± 1.5	2.31
**37-14**	22.4 ± 7.1	22.8 ± 8.5	-	25.9 ± 9.3	24.2 ± 4.4	1.07
**37-15**	29.7 ± 11.6	29.0 ± 5.4	-	21.8 ± 1.4	18.7 ± 5.2	1.17
**37-16**	20.0 ± 0.9	15.6 ± 2.6	-	14.8 ± 3.3	6.5 ± 2.2	2.28
**37-17**	14.1 ± 2.9	13.2 ± 5.7	-	6.8 ± 1.7	4.2 ± 1.1	1.62
**37-18**	16.5 ± 2.2	11.3 ± 7.9	-	5.4 ± 1.8	4.4 ± 0.8	1.23
**37-19**	25.6 ± 1.2	19.8 ± 4.6	-	12.1 ± 7.8	4.7 ± 1.4	2.57
**37-20**	>20	>20	-	>20	>20	-
**37-21**	>20	>20	-	>20	>20	-
**37-22**	>20	>20	-	>20	>20	-
**PXL**^2^ (nM)	4.8 ± 0.6	5.9 ± 1.9	8.4 ± 0.8	5.8 ± 0.2	2405.4 ± 44.8	-

^1^ Values are means ± standard deviation; ^2^ Paclitaxel (PXL; nM) was used as an experimental control.

**Table 4 molecules-24-02317-t004:** Cytotoxic activity data for actaline and napelline-type C_20_-diterpenoid alkaloids **55**~**63** and synthetic analogs **56-1**~**56-7** of luciculine.

	Cell Line/IC_50_ (μM) ^1^					
Alkaloids	A549	DU145	MDA-MB-231	MCF-7	KB	KB-VIN
Aconicarchamine A (**55**)	>40	-	>40	>40	>40	>40
Lucidusculine (**57**)	>20	>20	-	-	>20	>20
Flavadine (**58**)	>20	>20	-	-	>20	>20
12-Acetyllucidusculine (**59**)	>20	>20	-	-	>20	>20
1-Acetylluciculine (**60**)	>20	>20	-	-	>20	>20
Dehydrolucidusculine (**61**)	>20	>20	-	-	>20	>20
Dehydroluciculine (**62**)	>20	>20	-	-	>20	>20
12-Acetyldehydrolucidusculine (**63**)	>20	>20	-	-	>20	>20
**56-1**	>20	>20	-	-	>20	>20
**56-2**	>20	>20	-	-	>20	>20
**56-3**	>20	>20	-	-	>20	>20
**56-4**	>20	>20	-	-	>20	>20
**56-5**	23.3 ± 6.1	28.1 ± 11.1	-	-	31.8 ± 10.5	27.8 ± 1.9
**56-6**	>20	>20	-	-	>20	>20
**56-7**	>20	>20	-	-	>20	>20
**PXL**^2^ (nM)	4.8 ± 0.6	5.9 ± 1.9	8.4 ± 0.8	10.2 ± 0.9	5.8 ± 0.2	2405.4 ± 44.8

^1^ Values are means ± standard deviation; ^2^ Paclitaxel (PXL; nM) was used as an experimental control.

**Table 5 molecules-24-02317-t005:** Cytotoxic activity data for hetisine-type C_20_-diterpenoid alkaloids **64**~**67** and synthetic derivatives **67-1**~**67-19** of kobusine.

	Cell Line/IC_50_ (μM) ^1^						KB/KB-VIN Ratio
Alkaloids	A549	DU145	MDA-MB-231	MCF-7	KB	KB-VIN	
Ryosenamine (**64**)	>40	-	>40	>40	>40	>40	
9-Hydroxynominine (**65**)	>40	-	>40	>40	>40	>40	
Torokonine (**66**)	>40	-	>40	>40	>40	>40	
Kobusine (**67**)	>20	>20	-	-	>20	>20	
**67-1**	>20	>20	-	-	>20	>20	
**67-2**	>20	>20	-	-	>20	>20	
**67-3**	>20	>20	-	-	>20	>20	
**67-4**	>20	>20	-	-	>20	>20	
**67-5**	8.4 ± 1.4	9.3 ± 3.0	-	-	6.0 ± 0.8	7.5 ± 3.7	0.80
**67-6**	>20	>20	-	-	>20	>20	
**67-7**	6.7 ± 2.4	7.1 ± 2.0	-	-	5.3 ± 0.3	5.2 ± 1.2	1.02
**67-8**	19.5 ± 3.3	15.3 ± 5.6	-	-	13.9 ± 2.8	17.9 ± 1.8	0.78
**67-9**	>20	>20	-	-	>20	>20	
**67-10**	6.9 ± 1.7	7.0 ± 2.2	-	-	5.3 ± 0.6	5.5 ± 0.7	0.96
**67-11**	>20	>20	-	-	>20	>20	
**67-12**	>20	>20	-	-	>20	>20	
**67-13**	17.2 ± 0.9	13.2 ± 2.8	-	-	12.7 ± 1.1	14.1 ± 1.0	0.90
**67-14**	14.1 ± 0.7	9.6 ± 2.4	-	-	11.7 ± 0.6	10.9 ± 0.7	1.07
**67-15**	>20	>20	-	-	>20	>20	
**67-16**	>20	>20	-	-	>20	>20	
**67-17**	>20	>20	-	-	>20	>20	
**67-18**	8.1 ± 4.7	6.8 ± 2.0	-	-	5.2 ± 0.6	7.1 ± 2.6	0.73
**67-19**	5.5 ± 1.9	6.2 ± 3.1	-	-	4.1 ± 0.7	3.1 ± 1.6	1.32
**PXL**^2^ (nM)	4.8 ± 0.6	5.9 ± 1.9	8.4 ± 0.8	10.2 ± 0.9	5.8 ± 0.2	2405.4 ± 44.8	0.0067

^1^ Values are means ± standard deviation; ^2^ Paclitaxel (PXL; nM) was used as an experimental control.

**Table 6 molecules-24-02317-t006:** Cytotoxic activity data for hetisine-type C_20_-diterpenoid alkaloids pseudokobusine (**68**) and its synthetic analogs **68-1**~**68-37**.

	Cell Line/IC_50_ (μM) ^1^				KB/KB-VIN Ratio
Alkaloids	A549	DU145	KB	KB-VIN	
Pseudokobusine (**68**)	>20	>20	>20	>20	
**68-1**	>20	>20	>20	>20	
**68-2**	>20	>20	>20	>20	
**68-3**	>20	>20	>20	>20	
**68-4**	19.3 ± 4.5	15.3 ± 4.3	12.8 ± 1.7	10.2 ± 0.9	1.25
**68-5**	8.8 ± 4.5	7.6 ± 2.5	5.2 ± 1.3	6.3 ± 0.6	0.83
**68-6**	>20	>20	>20	>20	
**68-7**	15.4 ± 3.7	13.2 ± 2.0	11.1 ± 5.5	15.7 ± 1.5	0.70
**68-8**	>20	>20	>20	>20	
**68-9**	>20	>20	>20	>20	
**68-10**	8.0 ± 5.1	15.3 ± 2.9	14.9 ± 3.6	20.1 ± 13.5	0.74
15-Veratroylpseudokobusine (**68-11**)	>20	>20	>20	>20	
**68-12**	16.0 ± 5.5	16.9 ± 7.8	19.7 ± 3.1	14.7 ± 7.0	1.34
**68-13**	15.2 ± 6.4	16.6 ± 7.9	18.1 ± 4.3	12.2 ± 5.6	1.48
**68-14**	>20	>20	>20	>20	
**68-15**	5.8 ± 0.7	7.2 ± 1.9	6.4 ± 0.8	6.4 ± 1.8	1.00
**68-16**	>20	>20	>20	>20	
**68-17**	>20	>20	>20	>20	
**68-18**	>20	>20	>20	>20	
**68-19**	5.0 ± 1.1	5.2 ± 1.8	5.6 ± 1.2	5.6 ± 2.9	1.00
**68-20**	6.8 ± 0.7	7.7 ± 3.8	8.9 ± 3.7	6.2 ± 1.3	1.44
**68-21**	>20	>20	>20	>20	
**68-22**	17.9 ± 7.2	14.5 ± 7.2	15.7 ± 4.1	13.9 ± 3.3	1.13
**68-23**	>20	>20	>20	>20	
**68-24**	8.4 ± 1.7	6.5 ± 0.5	7.0 ± 1.3	6.4 ± 0.9	1.09
**68-25**	>20	>20	>20	>20	
**68-26**	>20	>20	>20	>20	
**68-27**	>20	>20	>20	>20	
**68-28**	>20	>20	>20	>20	
**68-29**	>20	>20	>20	>20	
**68-30**	>20	>20	>20	>20	
**68-31**	>20	>20	>20	>20	
**68-32**	6.4 ± 1.2	6.0 ± 3.3	6.6 ± 3.1	5.2 ± 1.0	1.27
**68-33**	>20	>20	>20	>20	
**68-34**	>20	>20	>20	>20	
**68-35**	>20	>20	>20	>20	
**68-36**	>20	>20	>20	>20	
**68-37**	>20	>20	>20	>20	
**PXL**^2^ (nM)	4.8 ± 0.6	5.9 ± 1.9	5.8 ± 0.2	2405.4 ± 44.8	

^1^ Values are means ± standard deviation; ^2^ Paclitaxel (PXL; nM) was used as an experimental control.

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
