# Peer review of "Cytotoxic Effects of Diterpenoid Alkaloids Against Human Cancer Cells"

_molecules, 2019, doi:10.3390/molecules24122317_

Reviewer 1 Report

Authors revealed the cytotoxic effects of isolated diterpenoid alkaloids against several human cancer cells including MDR subline KB-VIN. However some revisions should be done. 

There is no Materials and Methods section to explain what methods or what materials were used for the each experiment.

In line 88-89, it said, "These observations suggested that the alkaloid may be partially pumped out of KB-VIN cells by P-gp." Author should reveal the result in the result section not assuming something without evidence.

In line 230-231, it said, "All tested actaline and napelline-type C20-diterpenoid alkaloids were evaluated for 231 antiproliferative activity against four to five human tumor cell lines [24,26] (Table 4).". But the title of table 4 is "Cytotoxic activity data for diterpenoid alkaloids and derivatives". This is not just for table 4 but others. Unify the terms.

Discussion and conclusions section is a bit brief. Please discuss the common chemical characteristics of compounds which have anti-tumor activity or discuss the different chemical characteristics between cytotoxic compounds and non-cytotoxic compounds

Author Response

Reviewer #1: 

Authors revealed the cytotoxic effects of isolated diterpenoid alkaloids against several human cancer cells including MDR subline KB-VIN. However, some revisions should be done.

There is no Materials and Methods section to explain what methods or what materials were used for each experiment.

Authors’ Response:  The materials/compounds are included at the beginning of each section.Methods are given in the related references cited in this review of the authors’ extensive published work on diterpenoid alkaloids.

In lines 88-89, it said, "These observations suggested that the alkaloid may be partially pumped out of KB-VIN cells by P-gp." Author should reveal the result in the result section not assuming something without evidence.

Authors’ Response:  The noted sentence “These observations suggested that the alkaloid may be partially pumped out of KB-VIN cells by P-gp.” has been deleted.

In lines 230-231, it said, "All tested actaline and napelline-type C20-diterpenoid alkaloids were evaluated for antiproliferative activity against four to five human tumor cell lines [24,26] (Table 4).". But the title of table 4 is "Cytotoxic activity data for diterpenoid alkaloids and derivatives". This is not just for table 4 but others. Unify the terms.

Authors’ Response:  The titles of the structure and activity tables have been unified.  For Table 4. “Cytotoxic activity data for diterpenoid alkaloids and derivatives 124-138.” has been changed to “Cytotoxic activity data for actaline and napelline-type C20-diterpenoidalkaloids (55~63) and synthetic analogs (56-1~56-7)of luciculine.”.

Discussion and conclusions section is a bit brief. Please discuss the common chemical characteristics of compounds which have anti-tumor activity or discuss the different chemical characteristics between cytotoxic compounds and non-cytotoxic compounds

Authors’ Response:  We have added significantly more descriptions to the discussion section, as well as a figure describing the major active compounds and their distinguishing structural characteristics.

Reviewer 2 Report

Wada and Yamashita report on the cytotoxic effects of diterpenoid alkaloids against human cancer cells. Thus, the cytotoxic effects of considerable chemical library whit 199 natural or synthetic alkaloids were analyzed against human tumor cells [A549 (lung carcinoma), DU145 (prostate carcinoma), MDA-MB-231 (triple-negative breast cancer), MCF-7 (estrogen receptor-positive, HER2-negative breast cancer), KB (identical to cervical carcinoma HeLa derived AV-3 cell line), and multidrug-resistant (MDR) subline KB-VIN]. This article summarized the results obtained in these screening since 2012.  

-       The Discussion and Conclusions sections should appear separately.

-       Due to the large number of compounds tested in this article, an adequate discussion about structure-activity relationship should be included.

-       A detailed Experimental Section of how the biological analyzes were performed with the cell lines should be included.

-       In Tables 1 and 2 the column "average" is not significant, nor is it discussed in detail in the manuscript. It should be deleted.

-       The structures should appear in the manuscript numbered consecutively. Thus, in Figure 3 the numbers are not consecutive and should be get organized and re-numbered.  

-       Check 87 in Table 3.

 Author Response

Reviewer #2: 

Wada and Yamashita report on the cytotoxic effects of diterpenoid alkaloids against human cancer cells. Thus, the cytotoxic effects of considerable chemical library whit 199 natural or synthetic alkaloids were analyzed against human tumor cells [A549 (lung carcinoma), DU145 (prostate carcinoma), MDA-MB-231 (triple-negative breast cancer), MCF-7 (estrogen receptor-positive, HER2-negative breast cancer), KB (identical to cervical carcinoma HeLa derived AV-3 cell line), and multidrug-resistant (MDR) subline KB-VIN]. This article summarized the results obtained in these screening since 2012.

-       The Discussion and Conclusions sections should appear separately.

Authors’ Response:  We have separated these two sections. 

Line 331. “4. Discussion andConclusions” has been changed to “4. Discussion”. 

Line 385. “5. Conclusions” has been added.

Lines 386-387. “We have synthesized acylated derivatives of various C19- and C20-diterpenoid alkaloids. All alkaloids and their derivatives were screened against four to fivehuman tumor cell lines.” has been added.

-       Due to the large number of compounds tested in this article, an adequate discussion about structure-activity relationship should be included.

Authors’ Response:  We have added significantly more descriptions to the discussion section, as well as a figure describing the major active compounds and their distinguishing structural characteristics.

-       A detailed Experimental Section of how the biological analyzes were performed with the cell lines should be included.

Authors’ Response:  Methods are given in the related references cited in this review of the authors’ extensive published work on diterpenoid alkaloids.

-       In Tables 1 and 2 the column "average" is not significant, nor is it discussed in detail in the manuscript. It should be deleted.

Authors’ Response:  Tables 1 and 2have beenrevised as per the reviewer’s comment.

-       The structures should appear in the manuscript numbered consecutively. Thus, in Figure 3 the numbers are not consecutive and should be get organized and re-numbered.  

Authors’ Response:  Figure 3has beenrevised.

The numbers of compound were re-numberedas per the reviewer’s comment.

-       Check 87 in Table 3.

Authors’ Response:  Table 3has beenrevised.

Reviewer 3 Report

This paper is staggering in the number of compounds that are presented for the cytotoxicity in 5 tumor cell lines (199). As it is presented as a review, I was having trouble keeping sections straight, and compounds straight. It would be helpful if the authors cited both the natural products and synthesized products in a column adjacent to the structures in the figures. That would also help in grouping the compounds source. I am also having difficulty following the different sections, and would suggest a summary or conclusion at the end of the different alkaloid types (e.g., C18, then C19 etc). The final summary would then compare the relevant substitutions with activity from each class. Presenting this as a figure in the manner in which they were first shown would give a visual aid to the SAR that is suggested, leaving out the inactive compounds as this might cloud the discussion. It would be helpful to a reader who was using this as a guide for further work for the ATCC numbers be given for the different cell lines, perhaps at the beginning table with cell culture results shown. 

Authors should consider using underling of compound numbers rather than showing in parentheses. With the citations in brackets and the compound numbering in parens, it gets distracting. 

An average is shown for all the cell lines. This is not a relevant comparison. That should be omitted. 

So, a rethinking of the presentation of the information from the papers cited, revisions to the introduction as suggested in the manuscript attached, restructuring the introduction as it contains information that is not useful for this paper. 

Author Response

Reviewer #3: 

This paper is staggering in the number of compounds that are presented for the cytotoxicity in 5 tumor cell lines (199). As it is presented as a review, I was having trouble keeping sections straight, and compounds straight. 

It would be helpful if the authors cited both the natural products and synthesized products in a column adjacent to the structures in the figures. That would also help in grouping the compounds source.

Authors’ Response:  Natural alkaloid has been designated as“*” after the numbers of the alkaloids in the figures.

Figures 2-7. “*: Natural alkaloid” has been added.

I am also having difficulty following the different sections, and would suggest a summary or conclusion at the end of the different alkaloid types (e.g., C18, then C19 etc). The final summary would then compare the relevant substitutions with activity from each class. Presenting this as a figure in the manner in which they were first shown would give a visual aid to the SAR that is suggested, leaving out the inactive compounds as this might cloud the discussion. It would be helpful to a reader who was using this as a guide for further work for the ATCC numbers be given for the different cell lines, perhaps at the beginning table with cell culture results shown. 

Authors’ Response:  We have added a Figure 8 that shows the major active compounds and some of their distinguishing structural features.  Also, we have separated the discussion and conclusion section, as well as added more detail to the former section.

Authors should consider using underlining of compound numbers rather than showing in parentheses. With the citations in brackets and the compound numbering in parentheses, it gets distracting. 

Authors’ Response:  The compound numbers are in bold-face type, so we do not think that underlining them is necessary.

An average is shown for all the cell lines. This is not a relevant comparison. That should be omitted. 

Authors’ Response:  Tables 1 and 2have beenrevised as per the reviewer’s comment.

So, a rethinking of the presentation of the information from the papers cited, revisions to the introduction as suggested in the manuscript attached, restructuring the introduction as it contains information that is not useful for this paper. 

Authors’ Response:  We have made the following changes to the first part of the introduction as suggested by Reviewer #3 in the marked pdf file. 

Line 27. “primarily” has been deleted.

Line 28. “used” has been deleted.

Lines 28-29. “…others are alkylating agents, antimetabolites, anthracyclines, topoisomerase inhibitors, monoclonal antibodies, and other antitumor agents” has been deleted.

Lines 29-30. “Many Chemotherapeutic drugs affect cell division or DNA synthesis; therefore, many plant alkaloids also show similar functional properties, such as blocking cell division by preventing the normal functioning of microtubules.” has been changed to “Chemotherapeutic drugs that affect cell division by preventing the normal functioning of microtubules include the vinca alkaloids”.

Line 32. “(Ranunculaceae)” has been changed to “(Family Ranunculaceae)”.

Also from the PDF marked by Reviewer #3

Page, 1, line 35: “pyro, lactone more explanation needed?”

Authors’ Response:  These two terms have been further explained.  
Lines 34-35. pyro (C8=C15or C15=O), lactone (d-valerolactone rather than cyclopentyl C-ring).

Page 2, lines 54-58: (?) This statement might be the very reason that no one has looked seriously at this class of alkaloids for cytotoxicity. Indeed, Monk’s hood (aconitine) has been used in Europe from the times following the crusades topically because of the toxicity. It might be informative to include information on past uses and why this paper’s addressing those issues by modification of structures is important. 

Authors’ Response:  The following description has been added.

This extreme toxicity resulted in the use of Aconitumextracts as poisons in hunting and warfare [new reference 13], although extracts were also used as folkloric medicines by oral and topical routes. For example, the roots of Aconitumplants have been used as "bushi", an herbal drug in some prescriptions of traditional Japanese medicine for the treatment of hypometabolism, dysuria, cardiac weakness, chills, neuralgia, gout, and certain rheumatic diseases[reference 12 changed to reference 14]. However, proper processing is essential to reduce the content of toxic alkaloids and avoid inadvertent poisoning [new references 15-17]. Such obstacles encourage a good understanding of the relationships between structure and cytotoxic activity of aconitine and related compounds before they can be considered for modification and development as chemotherapeutic agents.

Reviewer 4 Report

The manuscript molecules-522566

fit the journal's aims and scopes and is related to a topic that can be of interest for readers.

Literature search on the general theme need to be revised, some reference related to the topic of natural anticancer compounds are missing and need to be cited. I suggest a revision of the first part of the introduction.

There are large number of auto-citation of authors that are in part related to their long experience on the specific topic and on the large number of considered compounds obtained in previous study. Some references also in this case are missing as example just for example

Diterpenoid alkaloids and phenol glycosides from Aconitum naviculare (Brühl) Stapf. Natural Product CommunicationsVolume 3, Issue 12, 2008, Pages 1985-1989 Molecules 17(5), pp. 5187-5194

The review is complete and many structures are presented. The main weakness is in my opinion the missing of some graph and general consideration that can at least in part help readers with not so much familiarity with this family of alkaloids to clearly understand which are the main key structural point related to the considered bioactivity.

Thus I suggest to add a general figure with schematic representation of the structure and highlight of point that induce strong changes in cytotoxic effect.

Also a general remark in the discussion/conclusion on the potential toxicity or in the presence in the literatyre of any selectivity test (as example on non tumor vs tumor tissues) or some data if present on toxicity in vivo model of some of the compound may be needed.

I enclosed pdf version with some comments.

Thus a more clear and schematic part for discussion should be prepared and the manuscript should be revised to be more clear to readed.

Author Response

Reviewer #4: 

The manuscript molecules-522566 fits the journal's aims and scopes and is related to a topic that can be of interest for readers.

From the PDF marked by Reviewer #4: The beginning of the abstract can be modified with more general sentence as. Diterpenoid alkaloids from plant of the genus....were isolated and classified...

Authors’ Response:  The first sentences of the Abstract have been changed as follows.
Diterpenoid alkaloids are isolated from plants of the genera AconitumDelphinium, and Garrya(Ranunculaceae)and classified according to their chemical structure as C18-, C19- and C20-diterpenoid alkaloids.The extreme toxicity of certain compounds, e.g., aconitine, prompts a thorough investigation of how structural features affect their bioactivities.

Literature search on the general theme need to be revised, some reference related to the topic of natural anticancer compounds are missing and need to be cited. I suggest a revision of the first part of the introduction.

From the PDF marked by Reviewer #4: The first nine references are quite old, and poorly related to general classes of natural anticancer compounds. I suggest adding a couple of more recent review on the topic. The authors can also refer to very highly cited review appeared in Journal of Natural Products or in Current medicinal Chemistry related to the anticancer natural products. I suggest but not exclusively to cite:

Nature: A vital source of leads for anticancer drug development 2009 PhytochemistryReviews 8(2), pp. 313-331 

Plant natural products in anticancer drug discovery 2010 Current Organic Chemistry14(16), pp. 1781-1791 

Pharmacological activities of natural triterpenoids and their therapeutic implications 2006 Natural Product Reports23(3), pp. 394-411 

Natural products as antimitotic agents 2014 Current Topics in Medicinal Chemistry 14(20), pp. 2272-2285 

Phytochemicals: Key to effective anticancer drugs 2019 Mini-Reviews in Organic Chemistry 16(2), pp. 141-158 

Authors’ Response:  The prior references 1-9 have been deleted and replaced with the above 5 references cited by Reviewer #4 as well as another 4 references published between 2010 and 2019.

There are large number of auto-citation of authors that are in part related to their long experience on the specific topic and on the large number of considered compounds obtained in previous study. Some references also in this case are missing as example just for example

Diterpenoid alkaloids and phenol glycosides from Aconitum naviculare (Brühl) Stapf. Natural Product Communications Volume 3, Issue 12, 2008, Pages 1985-1989 Molecules 17(5), pp. 5187-5194

Authors’ Response:  As mentioned by the reviewer, our intention in this review was to summarize our prior extensive research on diterpenoid alkaloids, which is why the above two cited studies by other groups were not included. We have now specified this distinction more clearly in the abstract as follows. “Therefore, our group has extensively studied natural diterpenoid alkaloids and semi-synthetic alkaloid derivativesover several decades, particularly for their cytotoxic effects against human tumor cells ……..”

The review is complete and many structures are presented. The main weakness is in my opinion the missing of some graph and general consideration that can at least in part help readers with not so much familiarity with this family of alkaloids to clearly understand which are the main key structural points related to the considered bioactivity. 

Thus, I suggest adding a general figure with schematic representation of the structure and highlight of point that induce strong changes in cytotoxic effect.

Authors’ Response:  We have added a Figure 8 that shows the major active compounds and some of their distinguishing structural features.  Also, we have separated the discussion and conclusion section, as well as added more detail to the former section.

Also, a general remark in the discussion/conclusion on the potential toxicity or in the presence in the literature of any selectivity test (as example on non tumor vs tumor tissues) or some data if present on toxicity in vivo model of some of the compound may be needed.

Thus, a more clear and schematic part for discussion should be prepared and the manuscript should be revised to be more clear to read.

Authors’ Response:  We added a description regarding the toxicity of Aconitumextracts to the Introduction. 

Lines 380-384. “Furthermore, previously our study, Antitumor properties and radiation-sensitizing effects of various types of novel derivatives prepared from C19- and C20-diterpenoidalkaloids were also investigated[19]. Two novel hetisine-type C20-diterpenoid derivatives (68-7and 68-20) showed significant suppressive effects against the Raji non-Hodgkin’s lymphoma cell line [20].” has been added.

Miscellaneous

Figure 2.“see structure below”has been added.

Table titles and Tables 1-6 have beenrevised.

Line 116. “[24,26]” has been changed to “[30,39]”.

Line 221. “(ratio of IC50KB / IC50KB-VIN: 1.55, 1.36, and 1.60, respectively)” has been changed to “(ratio of IC50KB / IC50KB-VIN: 1.55, 1.36, and 1.62, respectively)”.

Line 224. “and 1.24, respectively” has been changed to “and 1.23, respectively”.

Lines 246-247. “both the natural alkaloids (124126136-138) and synthetic analogs (127-132134135)” has been changed to “both the natural alkaloids (55, 57~63) and synthetic analogs (56-1~56-456-656-7)”.

Lines 249. “Because the related alkaloids 125129-132134135were” has been changed to “Because the related alkaloids 5760, 56-2~56-456-6, and 56-7were”.

Lines 251-252. “luciculine derivatives” has been changed to “luciculine (56)derivatives”.

Line 254; Figure 5 title. “Chemical structures of aconitine-type C19-diterpenoid alkaloids 124-138.” has been changed to “Chemical structures of actaline and napelline-type C20-diterpenoidalkaloids 55~63.”.

Line 266. “synthetic(143-146148151152153157-159)” has been changed to “synthetic (67-1~67-467-667-967-11,67-1267-15~67-17)”.

Line 266. “(IC50KB / IC50KB-VIN: 1.33)” has been changed to “(IC50KB / IC50KB-VIN: 1.32)”.

Lines 494-495; Ref. 39. “J. Nat. Med.2019, submitted.” has been changed to “J. Nat. Med.2019, DOI: 10.1007/s11418-019-01331-6.”.

Round  2

Reviewer 1 Report

The review article is well modified as reviewers' comments

Now it is good to be published 

Reviewer 2 Report

Dear editor,

All suggested changes have been introduced in the revised version

Sincerely

Reviewer 4 Report

The manuscript was implemented and comments to previous version were solved

Thus in the present form the manuscript can be accepted